# Evolution of Nanostructured Carbon Coatings Quality via RT-CVD and Their Tribological Behavior on Nodular Cast Iron

**Alejandra Moreno-Bárcenas** [1,2], **Jesus Alejandro Arizpe Zapata** [1], **Miguel Ángel Esneider Alcalá** [1], **Jaime Téllez Ramírez** [3], **Antonio Magaña Hernández** [3] and **Alejandra García-García** [1,*]

1   Grupo de Síntesis y Modificación de Nanoestructuras y Materiales Bidimensionales, Centro de Investigación en Materiales Avanzados S.C., Parque PIIT, Km 10, Autopista Monterrey-Aeropuerto, Apodaca 66628, Mexico; alejandra.moreno@cimav.edu.mx (A.M.-B); alejandro.arizpe@cimav.edu.mx (J.A.A.Z.); miguel.esneider@cimav.edu.mx (M.Á.E.A.)
2   Facultad de Ciencias Químicas, Universidad Autónoma de Nuevo León, San Nicolás de los Garza 64570, Mexico
3   R&D ARBOMEX S.A de C.V., Calle Norte 7 No.102, Cd. Industrial, Celaya, Guanajuato 38010, Mexico; ingfundi@arbomex.com.mx (J.T.R.); antonio.magana@arbomex.com.mx (A.M.H.)
*   Correspondence: alejandra.garcia@cimav.edu.mx

**Abstract:** One of the most critical problems in industry is the wear of materials. Graphene, as a tribological coating, has shown a tremendous impact on sliding surfaces. In this work, a few layers of graphene were grown on a nodular cast iron substrate, a material used in camshafts. The studied synthesis parameters in a rapid thermal chemical vapor deposition (CVD) furnace and the quality of the final coating are presented. The influence of hydrogen flow and cooling rate was evaluated, obtaining the best results in the few layers of graphene structure and deposition at 10 sccm and 4 °C/min. A standard ball-on-disk tribometer was used to assess the coefficient of friction on a few layers of graphene on nodular cast iron substrates. Laboratory test results show that the few layers of graphene coating resulted in a 60% reduction in coefficient of friction and close to a 70% reduction in volume removed versus the uncoated substrates. The surface of the substrate was not modified before a few layers of graphene growth to have a working surface close to camshafts obtained by the industrial process at ARBOMEX SA de CV.

**Keywords:** few layers graphene; graphene coating; friction coefficient; RT-CVD; ductile iron

## 1. Introduction

Today, the automotive industry is the largest and has had sustained growth in the last 50 years. This growth is expected to continue for the next five years, with the emergence of new technology and the development of new materials to construct power trains, making them more efficient and lighter. Efficiency is related to the percentage of energy supplied/harnessed use for the movement of the engine moving parts and is called thermal efficiency, which is between 10–12% since automobile engines have thermal losses of up to 72%. Of these, 3% correspond to heat losses due to the friction of moving surfaces that make up the engine. Friction, wear, and lubrication have been recognized to have the most significant influence on lifetime and machinery efficiency worldwide. In the 1960s, the so-called "Jost Report" was presented, which estimated the monetary savings that would have been made in the UK if industrial tribological solutions had been implemented. Nowadays, designers and manufacturers of machinery are making great efforts in development to meet the requirements to make more efficient vehicles, not only for economic reasons, but for the increasing pressure to comply with the reduction in $CO_2$ emissions into the atmosphere, which comes from the Kyoto Protocol [1]. Recently, different studies have been carried out to update the findings reported by the "Jost Report," in which they show energy losses due

to friction and wear in transportation, industrial, and public services, which reach up to 11% and can be saved by developing new tribological solutions [2].

Therefore, as new developments appear focused on the tribological behavior of the sliding parts, different approaches have emerged for the development of nanomaterials and two-dimensional materials as possible means of resolving the aspects mentioned above in the "Jost" reports. An important research segment focuses on the nanometric scale as a practical means of surface studying in contact and their tribological behavior in nanometric regimes.

Numerous efforts have been previously made, seeking to improve the tribological behavior of sliding surfaces using carbon nanotubes [3,4] and nano-flakes [5] as well as other materials, several of them based on 2D materials.

Among these two-dimensional materials, graphene, discovered around 2004 [6], has been extensively studied in various areas ranging from optics to its study as a tribological material. It has been proven to have the potential to meet the requirements to be usable in films as a solid lubricant, achieving friction coefficients as low as 0.05 [7].

However, a good solid lubricant would manage to be compatible with the substrate and its surface roughness, and capable of withstanding the stresses between the sliding surfaces. At this point, the concept of "tailored lubrication" is important since little information can be found in the literature thus far. Many studies are based on the growth of tribological material on commercially available substrates engineered or with preferential crystallographic orientation, which are not necessarily where the tribological material should be used. For graphene growth, several substrates have been studied. Different parameters have been proposed, though we found few studies that considered the substrate such as a polycrystalline surface different to copper and without simple pre-treatment before graphene growth.

Works such as reported by Khagendra et al. informed on the influence of graphene grown by CVD on a mirror-polished gray iron substrate, where the reduction in the coefficient of friction (COF) reached 53% compared to the original substrate without a coating [8]. As a solid lubricant, graphene contributes to forming an interface between the sliding surfaces, reducing nanomechanical processes, and reducing friction. However, little is known about the mechanical properties of the substrate after the CVD graphene growth processes. According to the iron–carbon diagram, the transformation temperature (Tc) is 727 °C; that is, the matrix can undergo changes since, to grow graphene, a temperature in the range of 800 to 1000 °C is necessary [9,10]. Therefore, the iron matrix will undergo recrystallization, which will have a detrimental effect on mechanical properties such as hardness and modulus of elasticity, among others that directly depend on the microstructure.

This research aimed to grow a few layers of graphene (FLG) on a nodular cast iron substrate with the roughness used industrially and to show the effects on the microstructure and the behavior of the COF. To undertake this, three experimental stages were designed. The first stage shows the impact of the hydrogen flow on the deposition of carbon atoms since it is related to methane dissociation, which is the principal responsibility to the conformation of the graphene network. The second stage consists of the implementation of the cooling ramps, aiming to preserve the properties of the nodular iron matrix and the quality graphene coating grew. The importance of protecting the microstructure of the matrix affects the mechanical properties and final performance of the component to be coated. The third part focuses on a method to maintain the most significant amount of the initial characteristics of the nodular cast iron matrix by developing a graphene-like growth below the Tc; this method ensures keeping the microstructure and the mechanical properties of the original substrate.

Therefore, the present study describes a promising methodology for manufacturing and developing a tailored tribological coating of FLG on nodular cast iron as a catalyst substrate, widely used in automotive frictional parts.

## 2. Materials and Methods

### 2.1. Few Layers of Graphene Synthesis

The FLG synthesis was carried out in a rapid thermal chemical vapor deposition furnace (RT-CVD) from Annealsys (Intercovamex, El Marquéz, Qro. México) with starting conditions from previous investigations with adjustments for a nodular iron substrate [10]. High purity nitrogen, hydrogen, and methane gases were used for the growth process at $1 \times 10^{-2}$ torr pressure. The substrates were nodular iron plates of 1.5 cm $\times$ 1.5 cm $\times$ 0.4 cm, produced in ARBOMEX S.A de C.V., a Mexican company located at Celaya and Apaseo Guanajuato, as well as El Salto Jalisco, Mexico that specializes in camshaft manufacturing. The initial microstructure of the substrates was mostly pearlitic according to SAE D7003 (ASTM 100-70-03) with free ferrite not greater than 10% and roughness > 500 nm. The chemical composition is as follows: %C—3.61, %Si—2.36, %Mn—0.83, %P—0.015, %S—0.008, %Mg—0.046, %V—0.008, %Cr—0.043, %Ni—0.103, %Al—0.013, %Cu—0.879, %Mo—0.03 with a balance iron and carbon equivalent (CE) of 4.40. The study was divided into three stages: the hydrogen effect on the FLG quality, the cooling velocity, and growth temperature influence.

The growth process was divided into four steps: heating, cleaning, growth, and cooling [11]. In the heating step, the temperature was increased up to 950 °C in a nitrogen atmosphere for five minutes. As the cleaning step begins, hydrogen gas is passed in a ratio of 1:100 with nitrogen. After 15 min, the growth step starts with the methane introduction at the reactor mixed with 1000 sccm of nitrogen and hydrogen for 15 min. Finally, the cooling step was carried out for 5 min from 950 °C to room temperature (RT). The growth time remained fixed in the hydrogen effect and cooling ramp studies.

### 2.2. Hydrogen Effect

The first stage of this study was the introduction of different hydrogen fluxes [12,13] corresponding to 0, 5, 10, 30, 50, 80, and 100 sccm at the growth step mixed with nitrogen (1000 sccm) and methane (10 sccm). The names of the obtained samples for each tested hydrogen flux were designated as R5-0H, R5-5H, R5-10H, R5-30H, R5-50H, R5-80H, and R5-100H, corresponding to 0, 5, 10, 30, 50, 80, and 100 sccm, respectively. The cooling rate from growth temperature until RT was the same in all experiments (190 °C/min).

### 2.3. Cooling Step Effect

The second stage was to study the cooling ramp as an influence on FLG growth. The ramp variation was considered from 950 °C to 700 °C for 10, 25, and 60 min. The resultant samples were labeled as R10-10H, R25-10H, and R60-10H.

### 2.4. Temperature Effect

The third stage was carried out at a temperature of 720 °C and two growth times (15 and 25 min). The samples were labeled <Tc-15M and <Tc-25M.

### 2.5. Mechanical Properties

2.5.1. Tribological Behavior

The tribology test, ball on disc, was carried out in accordance with the ASTM G99 standard. The equipment used was Standard Tribometer (Anton Paar, Houston, TX, USA). A chrome steel sphere (100Cr6) 6 mm in diameter was used for the ball and the disc was the iron substrate with the different treatments described above, the radius of wear was set at 3 mm, the sliding distance was 200 m, and the linear speed was 5 cm/s with a load of 5 Newtons, the friction coefficient was obtained from the test and with the help of an OLYMPUS GX-51 metallographic optical microscope (Olympus, Shinjuku, Tokyo, Japan) equipped with an OLYMPUS STREAM ESSENTIALS image analyzer (Olympus, Shinjuku,

Tokyo, Japan), it was possible to measure the width of the wear track, with which the volume lost due to friction could be calculated with the following equation, ASTM G99:

$$\text{Disk volume loss} = 2\pi R\left[r^2 sin^{-1}\left(\frac{d}{2r}\right) - \left(\frac{d}{4}\right)\left(4r^2 - d^2\right)^{\frac{1}{2}}\right] \tag{1}$$

where
  $R$ = wear track radius;
  $d$ = wear track width; and
  $r$ = ball end radius.

### 2.5.2. Hardness

The hardness was determined with a Brinell durometer DTLC 3000 (LECO, St Joseph, MI, USA) with a ball diameter of 10 mm, a conversion was made to HB, and it was related by employing tables with the tensile strength [14,15].

### 2.6. Structural Characterization

Raman measurements were carried out using a LabRAM HR Evolution Raman spectrometer (HORIBA, San Francisco, CA, USA). The different FLG coating Raman spectra were obtained with a wavelength of 633 nm using the mapping mode. The Raman spectra of coatings were taken in the substrate zones free of nodules. Microstructural properties were studied using electronic microscopes JEOL model JSM-6010 PLUS (JEOL, Akishima Tokyo, JP)/LA and FEI Nova NanoSEM 200 (Thermo Fisher Scientific Inc., Hillsboro, OR, USA) and colored by ImageJ software (University of Wisconsin at Madison, USA) to analyze the different coating zones.

## 3. Results and Discussion

### 3.1. Few Layers Graphene Synthesis

#### 3.1.1. Hydrogen Effect

The hydrogen flux effect on grown FLG quality on nodular iron was studied by Raman spectroscopy. The Raman spectrum of graphitic species contained three characteristic bands, D, G, and 2D, with G and 2D being the most prominent for graphene [16,17]. These bands can change their intensity ratio depending on the type of carbonaceous structure. The G band is associated with the $E_{2g}$ mode and attributed to a doubly degenerate zone center phonon [18]. D band is associated with induced defects in the graphitic network, and its width is directly proportional to the number of defects. On the other hand, the 2D band is associated with the number of layers, presenting a sharp and intense band for a monolayer [19,20].

The cast iron substrate without a coating was not perceptible as a response at laser excitation by Raman analysis, as can be observed in Figure 1a. This result was our background to compare the study after graphenic material growth. For R5-0, R5-5, R5-10, R5-30, and R5-50 samples, low D band intensity could be observed at ~1350 cm$^{-1}$. The above suggests that the low presence of defects in the graphene network showed a more significant ordering of the structure. The G band was located at 1592 cm$^{-1}$ where there was a sharp and intense peak for the samples from 5 until 50 sccm of hydrogen flux. The quality of the coating was evaluated by the $I_D/I_G$ ratio (see the graph of Figure 1b) where the sample with the lowest $I_D/I_G$ ratio value suggests a material with higher crystallinity. In our case, the sample with the lowest ratio was for the sample grown with 10 sccm of hydrogen.

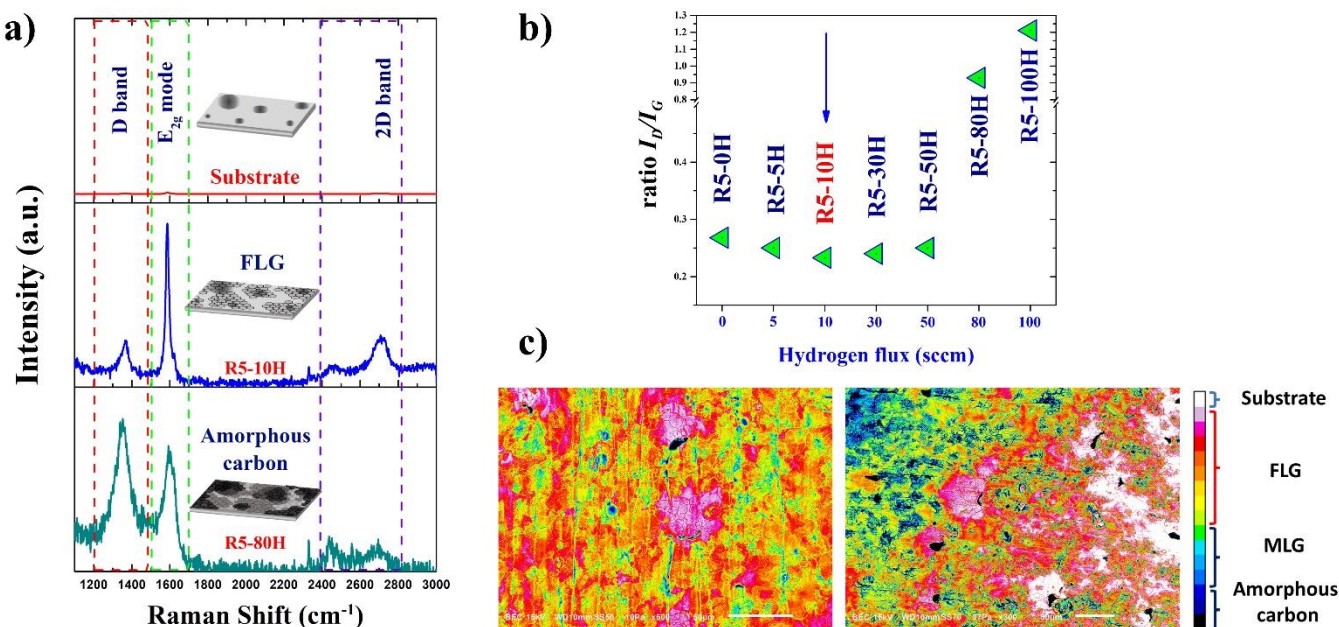

**Figure 1.** (**a**) Raman spectra of the uncoated substrate with a few layers of graphene (FLG) and amorphous carbon grown on nodular cast iron. (**b**) $I_D/I_G$ ratio versus hydrogen flux. The samples labeled in red were the best results highlighting the flow at 10 sccm, the samples labeled in green showed amorphous carbon formation at high hydrogen fluxes, and (**c**) colored SEM micrographs for R5-10H (**left**) and R5-80H (**right**) (bar corresponds to 50 μm).

An essential aspect of Raman analysis of graphene is the intensity of the 2D band, and it is related to the number of stacked graphene sheets (located at ~2700 cm$^{-1}$). The 2D band must be greater than the G band in a 2:1 ratio for monolayer graphene. Therefore, the structures obtained in the present study correspond to FLG, as shown in Figure 1a, with 10 sccm of hydrogen. This result is to be expected because the roughness of the substrate is an important factor that promotes the formation of multilayers [21]. This roughness factor is required for the piece's performance, so the presence of multilayers in both the ridges and in the valleys will become important in tribological tests.

The grown structures at 80 and 100 sccm of hydrogen presented a greater intensity of the D band concerning G, suggesting amorphous carbon formation by graphene layer stacking. This phenomenon was confirmed with a decreasing 2D band located at 2700 cm$^{-1}$ compared with the spectrum at 10 sccm of hydrogen, where a higher intensity in the 2D band was observed. In Figure 1b, a higher value for the $I_D/I_G$ ratio was observed as the hydrogen flux increased to 80 sccm; for this reason, from the hydrogen influence study, we decided to work with 10 sccm because it showed a higher crystallinity in FLG. Figure 1c shows the colored micrographs for 10 (left) and 80 (right) sccm of the hydrogen feed; these show different colors along the substrate correlated with the number of layers in the coating. In the sample at 10 sccm of hydrogen feed, a larger area with FLG (warm colors) was observed in all analyzed substrate zones than the 80 sccm feed, where amorphous carbon was evident. In the left superior corner of the micrograph at 80 sccm of hydrogen feed, we observed a greater concentration of amorphous material with the blue color, and at the center, we observed the presence of some FLG. The coating at the 80 sccm hydrogen feed showed heterogeneity and amorphous zones correlating to the Raman results.

### 3.1.2. Cooling Step Effect

The cooling effect on the coating's quality was studied by Raman spectroscopy. Figure 2a shows the Fe–C diagram, where the transformation temperature is indicated at 727 °C. The Raman spectrum (see Figure 2b) shows the vibration modes corresponding to induced disorder, crystallinity, and the number of layers in coatings on nodular cast

iron substrates. The cooling ramp in this study was considered from 950 °C to 700 °C. Figure 2c–e corresponds to the synthesis conditions of each test. The samples were labeled as R15-10H, R25-10H, and R60-10H for the 25, 10, and 4 °C/min cooling rates. An important aspect observed after spectrum obtention was the best quality obtained in the sample labeled as R60-10H, suggesting the best atomic restructure in the graphene network at long times of cooling. The ratio $I_{2D}/I_G$ was directly proportional to the cooling times increasing as follows: R60-10H > R25-10H.

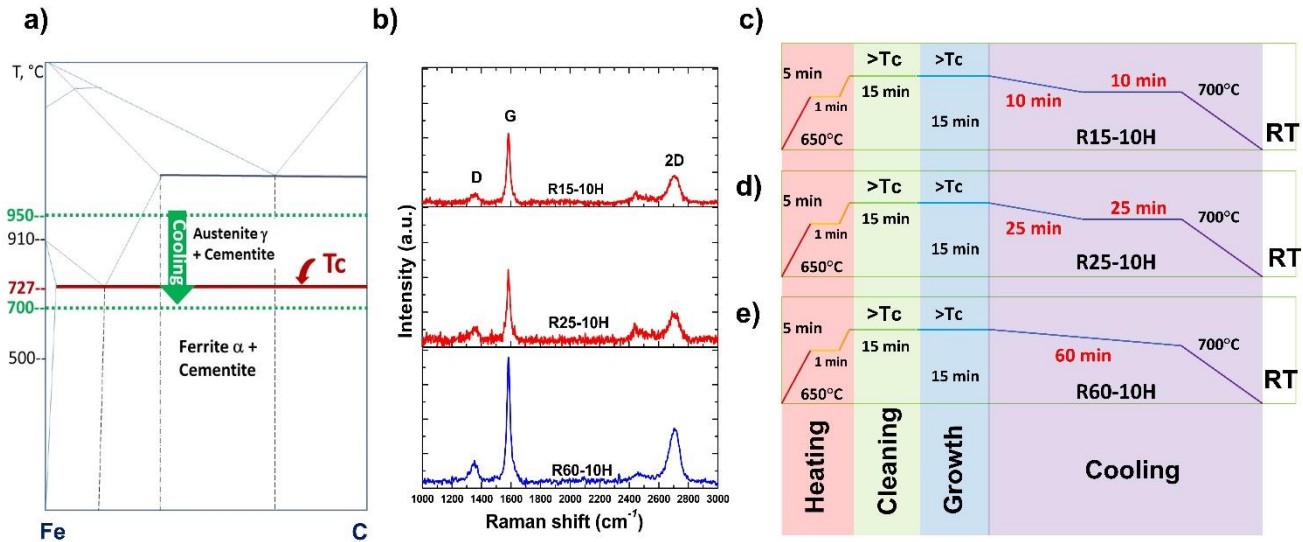

**Figure 2.** (**a**) Iron carbon phase diagram. (**b**) Raman spectrum of samples at different cooling rates, (**c**–**e**) cooling ramps for R15-10H, R25-10H, and R60-10H samples.

### 3.1.3. Temperature Effect

The results of this stage are shown in Figure 3, where the three characteristic bands for carbonaceous materials (D, G, and 2D) could be observed. The $I_D/I_G$ ratio indicates that the grown nanostructures contained defects in the carbon lattice. At the same time, the shape of the G band for the sample Tc-25M did not have a simple Lorentzian band as in the case of FLG (indicated by a red arrow), which suggests the formation of other carbon nanostructures. A small shoulder indicated by a red arrow was observed at ~1615 cm$^{-1}$, which was attributed to multi-walled carbon nanotubes (MWCNT) [22]. This peak was much less intense than the G peak and occurred by the second-order Raman scattering. The 2D band was less intense than the D band and was attributed to the second-order resonant mode, assigned to different carbon structures as two-dimensional sheets in graphene. The SEM micrographs confirmed this observation, as shown in Figure 3b,c, where carbon nanotubes of various lengths were observed. In Figure 3b, MWCNTs were observed with diameters from ~55–100 nm. Therefore, at temperatures below Tc, a mixture of carbon nanostructures was obtained where MWCNTs predominated. The zoom in micrograph 3c indicates that MWCNTs are open [23], this observation was confirmed by the low contrast indicated with a yellow arrow.

It has previously been reported that graphene growth is carried out through catalytic reaction where the substrate accomplishes the function of catalyst and support. To grow monolayer graphene on the substrate surface, carbon molecule adsorption, decomposition, and diffusion reactions must be carried out. However, these processes are not carried out in all metals under the same growth conditions. Monocrystalline copper (Cu) is the metallic substrate in which these processes are fulfilled and is known as the "self-limiting" effect [21,24]. On substrates other than Cu such as iron (Fe), graphene growth occurs through the diffusion of bulk carbon (C), attributed to the high solubility of C in Fe and its segregation during the cooling stage [22,25]. In the case of Cu(111), the graphene growth is continuous without the formation of islands; this is attributed to the negligible dissolution

and precipitation of C in Cu. In the case of the growth of graphene on nodular cast iron, and according to the obtained results in this work, a more significant dissolution of C in Fe is suggested compared with Cu, taking into consideration a precipitation process that can be controlled by the cooling stage and by the flow of injected hydrogen [23–28]. The role of hydrogen in this study was a reducing gas for oxidative species on the substrate surface and a diluent gas for the carbon precursor. At a hydrogen feed greater than 80 sccm, passivation of the surface occurred. The inactivation of nucleation sites gave way to carbon segregation and amorphous material formation. One important observation is the polycrystallinity in nodular cast iron compared with Cu(111). Nodular cast iron will grow a monolayer and graphene sheets over it, modulating the total number of layers by controlling the growth parameters. The best result at this work suggests a flow of 10 sccm of hydrogen feed and extended times for the cooling stage (>60 min from 950 to RT), these parameters promote the homogeneous formation of FLG before reaching the poisoning of the nucleation and passivation sites at the substrate surface. It is important to note that the substrate did not have a surface treatment before growth. The parameters reported in this work were based on the characteristics required to produce camshafts manufactured by the Arbomex company.

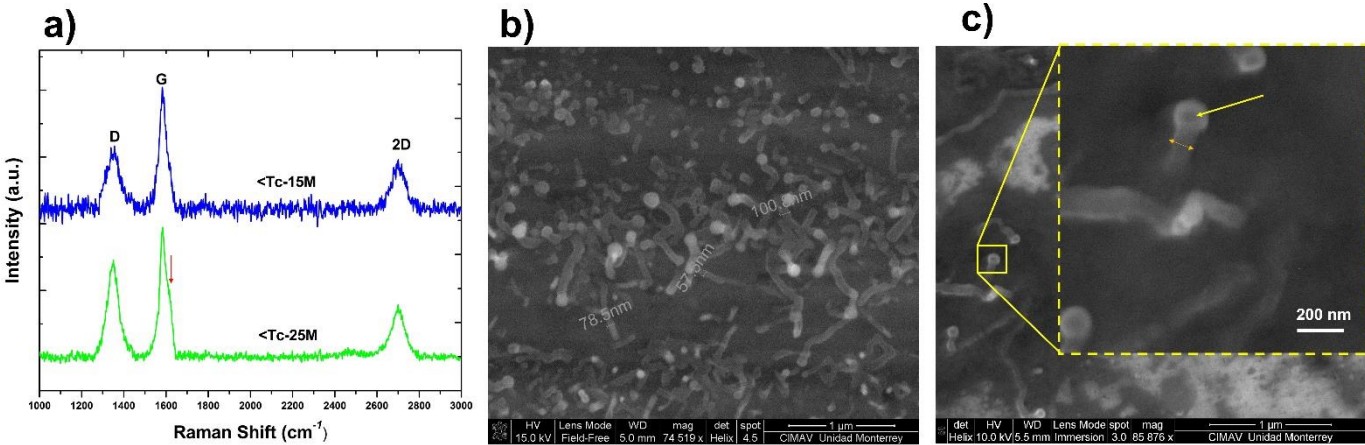

**Figure 3.** (**a**) Raman spectrum for growth temperatures lower than Tc with 15 and 25 min of methane injection. (**b**,**c**) SEM micrographs of grown carbon nanotubes on nodular cast iron.

### 3.2. Tribological Behavior

The COF for the samples at 10 and 80 sccm of hydrogen feed and the original substrate are shown in Figure 4d. The COF was lower for the sample with a coating grown with less hydrogen feed, correlating with the result obtained by Raman in which it was shown that the quality of the carbon material coating is important for a good performance and reduce friction. The reduction in the COF for the sample with 10 sccm was 20% less (see Figure 4c) than the bare substrate until the test reached 120 m. Beyond this value, the COF reached the value of the substrate (Figure 4d). When the graphene-type material was deposited with a concentration >80 sccm of hydrogen feed, COF was greater than the bare substrate from time zero, attributing this behavior to the amorphous carbon zones seen in the SEM images (Figure 1c). The difference in the COF in both samples was related to the number of layers present in the graphene-like material, denoting better tribological properties with FLG presence. The increase in COF for sample R5-10H after 120 mts can be attributed to the loss of FLG due to coupling lack between graphene islands. The uncoupling of FLGs could be attributed to the fast cooling ramp carried out in this sample and to the lack of junction between FLG with different orientations (see Figure 5). To ensure that the observed effects on COF value were due to FLG and carbon nanostructures, substrates were subjected to the growth process but without methane, and flux to avoid carbon deposition on the surface. The results are shown in Table 1 as R60-10H WG and <Tc WG.

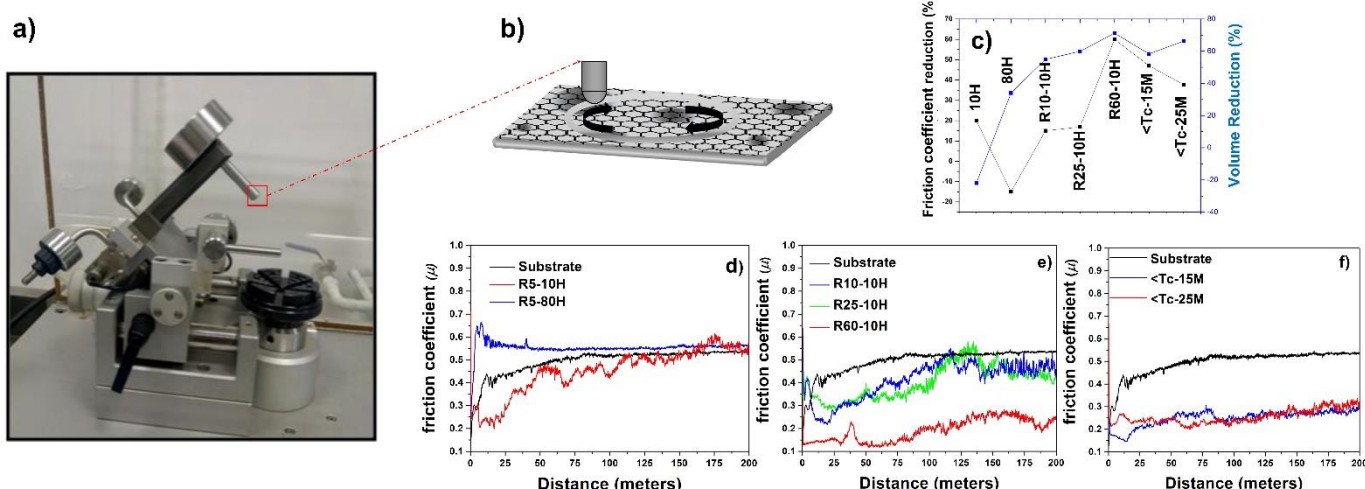

**Figure 4.** (**a**) Tribometer used to carry out the test, (**b**) test scheme on coated nodular cast iron substrates, (**c**) graphic of COF reduction percentage and volume removed reduction, (**d**) COF for R5-10H and R5-80H samples, (**e**) COF for R15-10H, R25-10H, and R60-10H, and (**f**) COF for the <Tc-15M and <Tc-25M samples.

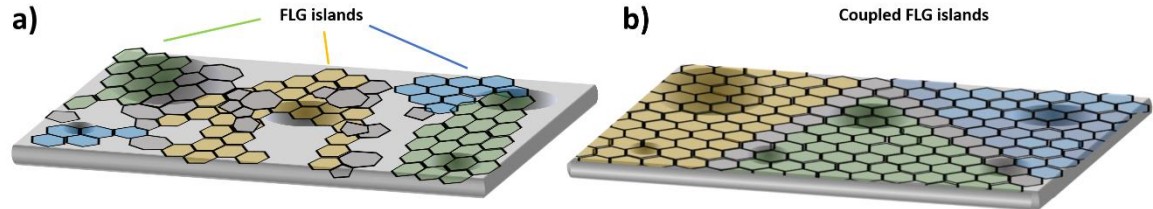

**Figure 5.** FLG scheme: (**a**) uncoupled FLG islands due to a fast-cooling ramp (190 °C/min) and (**b**) coupled FLG islands cooling at 4 °C/min.

**Table 1.** Friction coefficient and removed volume for the studied samples compared with the nodular cast iron substrate and hardness and tensile strength.

| Sample | Friction Coefficient μ | Removed Volume (mm$^3$) | HB | Ultimate Tensile Strength (Calculated) 1000 lb/in$^2$ |
|---|---|---|---|---|
| Substrate (as cast) | 0.53 | 0.2995 | 241 | 118 |
| R5-10H | 0.42 ± 0.17 | 0.3678 | 241 | 118 |
| R5-80H | 0.61 ± 0.06 | 0.1972 | 241 | 118 |
| R15-10H | 0.45 ± 0.13 | 0.1648 | 241 | 118 |
| R25-10H | 0.44 ± 0.12 | 0.1201 | 217 | 111 |
| R60-10H WG | 0.49 ± 0.01 | 0.2785 | 241 | 118 |
| R60-10H | 0.21 ± 0.08 | 0.0862 | 241 | 118 |
| <Tc WG | 0.40 ± 0.02 | 0.3323 | 217 | 111 |
| <Tc-15 M | 0.28 ± 0.2 | 0.1246 | 217 | 111 |
| <Tc-25 M | 0.33 ± 0.1 | 0.1001 | 217 | 111 |

Regarding the study with different cooling ramps, a decrease in the COF was observed for samples R15-10H and R25-10H. The COF decreased in the first 140 m test distance from 15 to 16.7% (see Figure 4e), with a more stable behavior than the R5-10H sample. Figure 4c shows a reduction in volume loss of 55 and 59.9%. These results correlate with the Raman spectra, showing that FLG has a good structural quality (Figure 2b). For sample R60-10H, a low COF was observed with a reduction of 60% in addition to a reduction due to the loss of volume of 71.2% (see Figure 4c), which indicates that the best parameters to obtain a FLG coating with a COF reduction is a low hydrogen feed (10 sccm), 950 °C, and slow cooling rate (4 °C/min).

Finally, the labeled samples <Tc-15M and <Tc-25M, which correspond to low-temperature graphene growth, showed results with a significant decrease in COF, a reduction of 47 and

37.7%, and a volume removed reduction of 58.3 and 66.5%, respectively, along with the entire test (200 m) (see Figure 4c,f). The aforementioned indicates that a mixture of carbonaceous nanostructures at the surface improves friction conditions. The studies for this finding on nodular cast iron substrates will be expanded in upcoming research. Table 1 summarizes the friction coefficients, loss of volume, hardness, and tensile strength for all of the carried out tests in the present study. The samples did not undergo a representative change in hardness and tensile strength, except for samples grown at temperatures below Tc. As can be seen, the COF values for the substrates subjected to treatment without FLG growth had a lower value than the original substrate. However, the COF for those with FLG or carbon nanostructures remained lower, indicating the contribution of these systems to the tribological properties.

The increase in COF for sample R5-10H after 120 M can be attributed to the loss of FLG due to the lack of linking between graphene islands that were separated from each other due to the absence of atomic ordering derived from a rapid cooling ramp. This is in contrast to what happened with a slow cooling ramp (60 min), where time was given for linking between the islands, improving the quality of the FLG. It is evident that COF decreases by carbonaceous nanostructures grown at low temperatures. This subject continues to be studied and will be reported as a continuation of the present work.

## 4. Conclusions

The present work found promissory parameters to grow tailor-made FLG on nodular cast iron without surface preparation. We found that excess hydrogen generated the growth of amorphous carbon, which in turn caused an increase in COF. Use of prolonged cooling ramps < 4 °C/min allowed for an improvement in the quality of the coating, in addition to maintaining the mechanical properties of the original substrate. The use of temperatures below the Tc enables the growth of a mixture of carbonaceous nanostructures, translating to a reduction in COF by 37–47%. However, the substrate hardness will be affected. The grown sample at 950 °C with a hydrogen feed of 10 sccm and a cooling rate of 4 °C/min reached a 60% reduction in the COF and 71.2% volume removed reduction concerning the original substrate. Combining the different growth conditions allowed us to obtain a FLG coating without altering the initial mechanical properties of nodular cast iron such as hardness, a critical parameter to scale this technique to components such as camshafts at an industrial level.

**Author Contributions:** Conceptualization, A.M.-B., A.G.-G. and J.T.R.; methodology, A.M.-B., A.G.-G.; formal analysis, A.M.-B., A.G.-G., J.A.A.Z. and M.Á.E.A.; investigation, A.G.-G.; resources, A.G.-G., J.T.R., A.M.H.; data curation, J.A.A.Z., M.Á.E.A.; writing—original draft preparation, A.M.-B.; writing—review and editing, A.M.-B., A.G.-G.; supervision, A.G.-G., J.T.R.; project administration, A.G.-G. All authors have read and agreed to the published version of the manuscript.

**Funding:** ARBOMEX SA DE CV by technological project with number 20474.

**Data Availability Statement:** The present study does not report any data.

**Acknowledgments:** The authors thank Mexican enterprise ARBOMEX SA de CV for the facilities given for the trial's development.

**Conflicts of Interest:** The authors declare no conflict of interest.

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
