# Peer review of "Evolution of Nanostructured Carbon Coatings Quality via RT-CVD and Their Tribological Behavior on Nodular Cast Iron"

_metals, doi:10.3390/met12030517_

Round 1

Reviewer 1 Report

The paper describes the optimisation of FLG coatings on a cast iron substrate to reduce friction and wear while minimising any reduction in hardness. The paper is clearly written with comprehensive references and the conclusions are justified by the results. The work is novel and suitable for the scope of the journal.

Author Response

Dear reviewer,

please see the attachment,

Reviewer 2 Report

Dear Authors!

Please, pay attention to the notions in the manuscript.

Use English editing service, because many sentences are hard to read, and some words seem like used mistakenly.

In the materials section, please, describe the tribological test correctly, especially the volume loss measurement. Add the analyses of friction surface.

Please, describe the properties of the material used as a substrate: chemical composition, initial microstructure, the effect of coating deposition on microstructure. Explain the methodology of surface roughness measurement and present the results. 

What was the weight gain of specimens after coating deposition?

You manifested the studies of microstructure and surface topography (lines 138-140), but I couldn't find this in the text.

Improve the figures: it is very hard to read the words and digits on them.

Update the reference list and add missing references.

I recommend You to do the Major revision of the manuscript.

Author Response

(The authors gave the same response as above.)

Reviewer 3 Report

Graphene is a unique allotropic form of carbon. Graphene sheets are considered as nanoparticles according to the dimensions (the width of the sheet is in between 1 – 100nm range). The carbon atoms of this sheet have four bonds including three sigma bonds around a carbon atom and one pi bond oriented out of the plane. This is a 2D monatomic layer in which carbon atoms are arranged in hexagonal arrangements. Graphene in its properties is a metal-like form of carbon with high electrical and thermal conductive properties. Layers of carbon obtained by CVD technique and having a thickness of the order of tens nm are not graphene. According to their properties, they are amorphous graphite. However, in the literature of recent decades, they are arbitrarily referred as graphene with more or less defects in the crystal structure. This is due to the desire to draw special attention of readers to obtained results. This article is no different in this regard.

The title of the article contains the combination "tailored graphene". However, the article investigates the effect of various carbon island RT-CVD structures on the frictional properties of cast iron. Among them is a mixture of defective graphene with crystalline multilayer graphite, nanotubes, and amorphous carbon. It is more correct to formulate the title of the article “Evolution of carbon coatings quality via RT-CVD and its tribological behavior on coated nodular cast iron”

The purpose of the article is to prove that applying a solid lubricant in the form of a carbon coating to cast iron will improve its frictional properties. This is a perfectly reasonable assumption, since graphite in cast iron acts as a solid lubricant. When applying a carbon coating, cast iron is subjected to additional heat treatment. During cooling from 950°C to 700°C graphite content increases and it increases with decreasing cooling rate. Thus, the highest graphite content was in sample R60-10H compared to other samples, since it was cooled at the lowest rate of 4 °C/min. Of course, this reduces the coefficient of friction and improves the antifriction properties of cast iron. Regardless of whether there are islands of the so-called graphene on its surface.

  1. For these reasons, in Table 1 it is necessary to give the coefficients of friction for samples of cast iron after additional heat treatment with a cooling rate of 4 °C/min. Without this comparison, it is impossible to draw a conclusion about the positive role of the coating in the wear resistance of cast iron.
  2. How can the authors explain the complete lack of correlation between the change in the friction coefficient and the removed volume according to the results presented in Table 1
  3. On line 81 before FLG add a «few layers of grapheme».
  4. On Lines 214-215 “Therefore, at temperatures below Tc, a mixture of 214 carbon nanostructures is obtained where multiwalled nanotubes predominate. Please provide evidence for the existence of multilayer nanotubes.

Author Response

(The authors gave the same response as above.)

Round 2

Reviewer 2 Report

Thank You for the work. Now manuscript looks much better, and I think it may be published in present form